# DPQuant: Efficient and Differentially-Private Model Training via Dynamic Quantization Scheduling

**Yubo Gao**
University of Toronto
Toronto, Ontario
ybgao@cs.toronto.edu

**Renbo Tu**
University of Toronto
Toronto, Ontario
renbo.tu@mail.utoronto.ca

**Gennady Pekhimenko**
University of Toronto
Toronto, Ontario
pekhimenko@cs.toronto.edu

**Nandita Vijaykumar**
University of Toronto
Toronto, Ontario
nandita@cs.toronto.edu

## Abstract

Differentially-Private SGD (DP-SGD) and its adaptive variant DP-Adam are powerful techniques to protect user privacy when using sensitive data to train neural networks. During training, converting model weights and activations into low-precision formats, i.e., *quantization*, can drastically reduce training times, energy consumption, and cost, and is thus a widely used technique. In this work, we demonstrate for the first time that quantization causes significantly higher accuracy degradation in DP training compared to regular SGD. We observe that this is caused by noise injection, which amplifies quantization variance, leading to disproportionately large accuracy degradation. To address this challenge, we present DPQuant, a dynamic quantization framework that adaptively selects a changing subset of layers to quantize at each epoch. Our method combines two key ideas that effectively reduce quantization variance: (i) *probabilistic sampling* that rotates which layers are quantized every epoch, and (ii) *loss-aware layer prioritization*, which uses a differentially private loss sensitivity estimator to identify layers that can be quantized with minimal impact on model quality. This estimator consumes a negligible fraction of the overall privacy budget, preserving DP guarantees. Empirical evaluations on ResNet18, ResNet50, and DenseNet121 across a range of datasets demonstrate that DPQuant consistently outperforms static quantization baselines, achieving near Pareto-optimal accuracy-compute trade-offs and up to $2.21\times$ theoretical throughput improvements on low-precision hardware, with less than $2\%$ drop in validation accuracy. We further show that our framework extends to DP-Adam with similar gains.

## 1 Introduction

Differentially Private Stochastic Gradient Descent (DP-SGD) (Abadi et al., 2016) enables training neural networks on sensitive data while providing formal privacy guarantees. To improve the efficiency of such training on modern hardware, the use of *low-precision* arithmetic and data formats, i.e., *quantization*, has gained widespread interest (Gholami et al., 2021; Jacob et al., 2017). Quantization can significantly reduce the amount of computation and memory required, thus reducing the latencies, cost, and energy consumption during training and inference, often with little to no loss in model accuracy (Micikevicius et al., 2022). These benefits are especially important in resource-constrained settings such as federated learning with edge devices, where support for full-precision arithmetic is limited and compute budgets are constrained.

Modern accelerators, ranging from datacenter GPUs to mobile NPUs, are rapidly adopting *ultra-low precision formats* such as FP8, INT4, or FP4. NVIDIA Blackwell (NVIDIA Corporation, 2024) provides $4\times$ throughput for FP4 matmuls compared to FP16; AMD Instinct GPUs, Google TPUs

and AWS Trainium support FP8 (Advanced Micro Devices, Inc., 2023; 2025; Vahdat, 2025; Amazon Web Services, 2025), and Qualcomm Hexagon supports INT4/INT8 (Qualcomm Technologies, Inc., 2024). Leveraging these compute capabilities in model training would enable significant performance and scalability improvements.

In this work, we observe that applying low-precision quantization directly to DP training often leads to *significant accuracy degradation*, as severe as a $40\%$ drop. While non-DP training is typically robust to quantization, the gradient clipping and noise addition steps in DP-SGD/DP-Adam interact poorly with low-precision arithmetic leading to poor convergence as explained in Section 4.

Our goal is to develop an automatic mechanism to effectively quantize DP training, while minimizing the impact on model accuracy and the privacy budget. We observe that *quantizing only a subset of layers* and *selectively varying this subset every epoch* can preserve most of the efficiency gains from quantization while maintaining model accuracy. DPQUANT uses two core techniques that are implemented in a *differentially private* framework for dynamic quantization scheduling:

1. **Probabilistic layer sampling**, which rotates which layers are quantized every epoch to distribute quantization variance across the network, decreasing overall quantization variance;

2. **Loss-aware prioritization**, which uses a loss sensitivity estimator to selectively quantize layers that have minimal impact on model accuracy.

Specifically, in this work we make the following contributions:

1. We are the first to demonstrate and explain the significant accuracy degradation that arises when employing ultra-low-precision (FP4/INT4) quantization with DP training (DP-SGD and DP-Adam), compared to the mild impact observed in non-private SGD and in 16-bit DP training.

2. We introduce DPQUANT, a differentially private lightweight mechanism that minimizes quantization-induced loss by (a) probabilistically sampling which layers to quantize every epoch and (b) prioritizing layers with lower sensitivity—while incurring only a negligible cost to the overall privacy budget.

3. We demonstrate that DPQUANT achieves near Pareto-optimal accuracy-speed tradeoffs across a range of compute and privacy budgets, outperforming static (fixed-layer) quantization baselines.

## 2 RELATED WORKS

**Post-training quantization (PTQ)** (Banner et al., 2019; Jacob et al., 2017; Nagel et al., 2021) aims to accelerate inference through low-precision computations. A neural network is first trained in full-precision and then its weights are quantized. The conversion to quantized formats typically involves a small calibration dataset (Hubara et al., 2021; Nagel et al., 2019) to allocate quantization bit-widths in different parts of the model and to perform bias correction. PTQ methods are orthogonal to this work since they do not optimize training.

**Quantization-aware training (QAT)** (Krishnamoorthi, 2018) ameliorates the aforementioned accuracy loss by training in lower precision. However, QAT requires additional overhead such as hardware simulation (Wang et al., 2019), sensitivity estimation (Dong et al., 2019), and quantizer tuning (Esser et al., 2020; Sakr et al., 2022), which typically cancels out raw bit-width speedups, making QAT suited for accelerating inference but not training (Chen et al., 2024).

**Gradient compression** (Lin et al., 2020; Alistarh et al., 2017; Alimohammadi et al., 2023; Wen et al., 2017; Shi et al., 2020) reduces communication costs by compressing gradients in distributed settings, either through sparsification (Stich et al., 2018; Yu et al., 2017) or low-rank approximation (Vogels et al., 2019; Idelbayev and Carreira-Perpinán, 2020). Notably, Youn et al. (2023) combine quantization and the noising mechanism to achieve differential privacy while reducing communication. However, compression does not lower the arithmetic cost of training. In addition, these methods often rely on assumptions such as full gradient availability or error feedback accumulation (Karimireddy et al., 2019), which are difficult to satisfy under DP constraints.

**Mixed-precision training** (Choi et al., 2018; Zhou et al., 2018; Sun et al., 2020; Chmiel et al., 2024; Micikevicius et al., 2022) aims to reduce training cost by operating on lower-precision data types (e.g.,

FP16, BF16, FP8, and FP4). Prior work has shown that 16-bit mixed-precision training (FP16/BF16) is compatible with DP-SGD with little accuracy loss (Li et al., 2022; Bu et al., 2023). However, pushing to ultra-low precision (FP4/INT4) under DP remains challenging: we show in Section 4 that DP noise amplifies quantization variance at these bit-widths, causing significant accuracy degradation that does not arise at 16-bit precision.

**Quantization as a privacy mechanism.** Several works use quantization to provide or strengthen DP guarantees in federated or decentralized settings (Kang et al., 2024; Kim et al., 2022; Wang and Basar, 2022; Xiong et al., 2016). In these approaches, quantization is applied *after* gradients are computed in full precision, so there is no effect on subsequent forward/backward passes. DPQUANT instead quantizes activations and weights *during* training to reduce arithmetic cost.

**Improved DP mechanisms.** A complementary line of work improves the DP algorithm itself for better utility at a given $\varepsilon$, whether through modified optimizers (Wang et al., 2024; Wei et al., 2022; Park et al., 2023; Denisov et al., 2023) or by reducing the trainable parameter space via pruning or layer selection (Adamczewski and Park, 2023). These methods are orthogonal to DPQUANT, which keeps the full dense model and instead optimizes how much computation can run in low precision.

**Gradient clipping optimizations for DP-SGD.** (Li et al., 2022; Bu et al., 2022; Lee and Kifer, 2020; Subramani et al., 2021) optimize the per-sample gradient clipping for DP-SGD by eliminating redundant computation and increasing vectorized computation, leading to better hardware utilization. These works are orthogonal to DPQUANT.

## 3 PRELIMINARIES

### 3.1 DIFFERENTIALLY-PRIVATE DNN TRAINING

We first recall the standard definition of differential privacy:

**Definition 1** (Differential Privacy, (Dwork and Roth, 2014)). *A randomized algorithm $\mathcal{A}$ satisfies $(\varepsilon, \delta)$-differential privacy if for all adjacent datasets $D, D'$ differing on at most one example, and for all measurable sets $S$ in the output space,*

$$\Pr[\mathcal{A}(D) \in S] \leq e^\varepsilon \Pr[\mathcal{A}(D') \in S] + \delta.$$

**Definition 2** (DP-SGD, (Abadi et al., 2016)). *Differentially Private Stochastic Gradient Descent (DP-SGD) is a variant of SGD that satisfies $(\varepsilon, \delta)$-differential privacy by clipping and perturbing per-example gradients. At each iteration $t$, the update rule is:*

$$\theta_{t+1} \leftarrow \theta_t - \eta \left( \frac{1}{|B|} \sum_{i \in B} \text{clip}_C(\nabla \mathcal{L}(\theta_t, x_i)) + \mathcal{N}(0, \sigma^2 C^2 \mathbb{1}) \right),$$

*where $B$ is a minibatch of training examples, $\text{clip}_C(\cdot)$ scales the gradient to have $\ell_2$ norm at most $C$, and $\mathcal{N}(0, \sigma^2 C^2 \mathbb{1})$ is Gaussian noise added to ensure privacy.*

### 3.2 QUANTIZATION AND MIXED PRECISION TRAINING

Modern hardware accelerators such as NVIDIA GPUs, Google TPUs, and Qualcomm Hexagon NPUs provide dedicated support for low-precision arithmetic, including fp16, bfloat16, and increasingly lower bitwidth formats like fp8, fp6, and fp4. These formats enable faster computation by reducing arithmetic complexity, memory footprint, and data transfer costs.

While prior work(Chmiel et al., 2024) has demonstrated that full training in low-bit formats (e.g., fp4) can often retain accuracy under standard SGD, extending these techniques to differentially private training remains challenging. The clipping and noise injection steps in DP-SGD amplify quantization errors and increase gradient variance, making DP training more sensitive to precision loss. Fully quantized DP-SGD thus often results in severe degradation unless carefully tuned.

## 4 DEGRADATION OF DP TRAINING FROM QUANTIZATION

Figure 1 presents a case study of ResNet18 trained on GTSRB, where the forward and backward convolution operators are quantized to FP4 to evaluate the effects on training. Figure 1a shows the

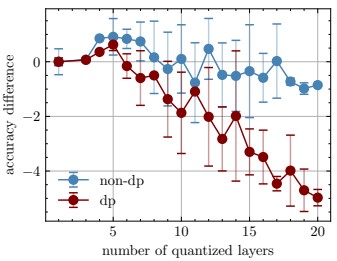 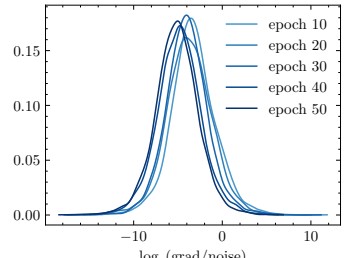 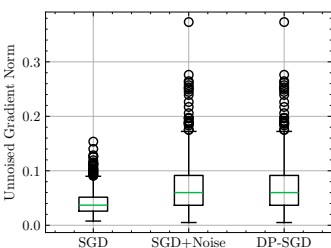

(a) Accuracy loss due to quantization between DP and non-DP SGD

(b) Distribution of gradient/noise ratios in `conv1` of ResNet18 – Gradient mostly dominated by noise

(c) Distribution of gradient norms between SGD, SGD and only noise injection, and full DP-SGD

Figure 1: Comparing quantized SGD vs DP-SGD ResNet18 training on the German Traffic Sign Recognition Benchmark (GTSRB) (Stallkamp et al., 2011) dataset

accuracy loss compared to the non-quantized baseline for different degrees of quantization (in terms of the number of layers quantized), and the error bars represent the results when different subsets of layers are chosen for quantization, both for DP and non-DP training. For the non-DP SGD baseline, fully-quantized training results in only a modest accuracy drop of around 1%. In contrast, DP-SGD experiences a much greater degradation, up to 5%. Furthermore, the variance in performance due to different layers being quantized is substantially higher under DP-SGD. We observe similar trends in other neural networks and datasets (included in Appendix A.6).

We hypothesize that the increased sensitivity to quantization can be attributed to noise injection in DP-SGD as follows. In iteration $t$ in the DP-SGD training, the gradients $\mathbf{g}_t$ are first clipped to obtain $\bar{\mathbf{g}}_t$ where $||\bar{\mathbf{g}}_t||_2 \leq C$ (section 3.1). Next, noise $\mathbf{n}_t \sim \mathcal{N}(0, \sigma^2 C^2 \mathbb{1})$ is sampled, and finally the weights are updated using the sum of the noise and clipped gradient:

$$\mathbf{w}_{t+1} = \mathbf{w}_t - \eta \left( \bar{\mathbf{g}}_t + \mathbf{n}_t \right) \tag{1}$$

We assume the noise scale is $\sigma \in (0.5, 10)$ (Abadi et al., 2016; De et al., 2022), in line with common configurations reported in the DP-SGD literature. Since the standard deviation of the injected noise $\mathbf{n}_t$ is equal to the 2-norm of the clipped gradients, the $\infty$-norm of the noise $\mathbf{n}_t$ (i.e. its largest component) is roughly on the same order as $||\mathbf{g}||_2$. Since in higher dimensions $||\bar{\mathbf{g}}||_2 \gg ||\bar{\mathbf{g}}||_\infty$ due to the 2-norm growing much faster than the $\infty$-norm, combining we have:

$$||\mathbf{n}_t||_\infty \approx ||\bar{\mathbf{g}}_t||_2 \gg ||\bar{\mathbf{g}}_t||_\infty \tag{2}$$

This relation is also demonstrated empirically in Figure 1b, where on average the magnitude of the clipped gradient elements of $\bar{\mathbf{g}}$ is $2^5$ times smaller than that of the injected noise $\mathbf{n}$.

The weight update (Equation 1) with noisy gradient updates amplify the norms of raw gradients in subsequent iterations. To show this, we first write the weight update as:

$$\Delta\mathbf{w}_t = \mathbf{w}_{t+1} - \mathbf{w}_t = -\eta \left( \bar{\mathbf{g}}_t + \mathbf{n}_t \right) \tag{3}$$

The weight update $\Delta\mathbf{w}_t \approx \eta \, \mathbf{n}_t$ due to $\mathbf{n}_t$ being much larger in norm than $\bar{\mathbf{g}}_t$, and thus:

$$||\Delta\mathbf{w}_t||_\infty \approx \eta \, ||\mathbf{n}_t||_\infty = \mathcal{O}(||\bar{\mathbf{g}}_t||_2) \tag{4}$$

where the asymptotic equality holds due to Equation 2. Assuming $L$-Lipschitz-smoothness of the loss with respect to the gradients:

$$||\mathbf{g}_{t+1} - \mathbf{g}_t||_\infty \leq L \, ||\mathbf{w}_{t+1} - \mathbf{w}_t||_\infty = L \, ||\Delta\mathbf{w}_t||_\infty \tag{5}$$

We now show that the $\infty$-norm of the raw gradients (i.e. before clipping and noising) of the next iteration is bounded by $||\bar{\mathbf{g}}_t||_2$ using the inverse triangle inequality with Equation 4 and 5:

$$||\mathbf{g}_{t+1}||_\infty \geq ||\mathbf{g}_{t+1} - \mathbf{g}_t||_\infty - ||\mathbf{g}_t||_\infty = \mathcal{O}(||\bar{\mathbf{g}}_t||_2) \tag{6}$$

Equation 6 shows that elements of the raw gradients in the next iteration are bounded by the much larger $\mathcal{O}(||\bar{\mathbf{g}}_t||_2)$ rather than the usual $\mathcal{O}(||\bar{\mathbf{g}}||_\infty)$ in normal SGD. As a result, we expect elements of the raw gradients in DP-SGD to be larger in magnitude than in non-DP training.

Notably, the batch size has a negligible effect on norms, even though a larger batch size leads to a smaller variance of the stochastic gradients. Therefore, we omit it in this analysis; we include further discussions and empirical evaluations of this in appendix A.1.

To show this empirically, we plot the norms of intermediate gradients under both SGD and DP-SGD using the same hyperparameters in Figure 1c, where the DP-SGD intermediate raw gradients are $2\times$ larger in the average and worst case. This phenomenon has also been observed in prior work (Du et al., 2022), showing an even larger gap than what we observe in gradient norms later in training.

We now demonstrate that the much larger raw gradients under DP-SGD result in much higher quantization variance.

**Proposition 1.** *Let $q : \mathbb{R}^n \to \mathbb{R}^n$ be an unbiased (i.e. $\mathbb{E}[q(\mathbf{x})] = \mathbf{x}$) and scale invariant (i.e. $q(\lambda \mathbf{x}) = \lambda q(\mathbf{x})$) quantizer. Assume $q(\mathbf{x})$ quantizes values onto some finite grid. Let $\mathbf{x}$ be sampled from an absolutely continuous distribution. Then the quantizer variance $\mathrm{Var}(q(\mathbf{x})) = \Theta(||\mathbf{x}||_\infty^2)$. Proof: See Appendix A.10.*

Using Prop. 1, we can more precisely express the variance of the quantization as follows:

$$\text{(under DP-SGD)} \quad \mathrm{Var}\big(q(\mathbf{g}_{t+1}) \,|\, \mathbf{g}_{t+1}\big) = \mathcal{O}\big(||\mathbf{g}_{t+1}||_\infty^2\big) = \mathcal{O}\big(||\mathbf{g}_t||_2^2\big)$$

$$\text{(under SGD)} \quad \mathrm{Var}\big(q(\mathbf{g}_{t+1}) \,|\, \mathbf{g}_{t+1}\big) = \mathcal{O}\big(||\mathbf{g}_t||_\infty^2\big)$$

The quantization variance above is *in addition* to the existing variance of the stochastic gradients, as well as noise injected by DP-SGD. In higher dimensions, $||\mathbf{g}_t||_2 \gg ||\mathbf{g}_t||_\infty$, quantization contributes much more variance to the gradients, hence leads to slower and less reliable convergence (Johnson and Zhang, 2013) and accuracy degradation. To address this challenge, we aim to reduce the quantization-induced variance.

**Extension to DP-Adam/DP-AdamW.** The above argument for DP-SGD relies on DP noise dominating the raw gradients, which leads to disproportionate quantization variance. In DP-Adam, noise is added to the clipped gradients first, and Adam's preconditioner then rescales the noisy gradients coordinate-wise. Since the preconditioner applies the same scaling to both signal and noise, it preserves the relative scale between them, so quantized DP-Adam incurs the same quantization overhead as DP-SGD.

We can reason about this more directly by considering the per-coordinate signal-to-noise ratio. In DP-Adam, each noisy gradient coordinate is $\tilde{g}_i = g_i + n_i$, where $g_i$ is the true gradient and $n_i \sim \mathcal{N}(0, \sigma^2 C^2)$. The preconditioner divides by a running second-moment estimate, so at stationarity the preconditioned coordinate is $u_i \approx (g_i + n_i)/\sqrt{g_i^2 + \sigma^2 C^2}$ with $\mathrm{Var}(u_i) \leq 1$.

In the strong DP regime (where $\sigma C \gg ||\mathbf{g}||_2 \gg g_i$), the mean (i.e. signal) is:

$$|\mathbb{E}[u_i]| = \frac{|g_i|}{\sqrt{g_i^2 + \sigma^2 C^2}} \approx \frac{|g_i|}{\sigma C} \leq \frac{1}{\sigma} \tag{7}$$

where the last inequality is due to $|g_i| \leq ||\mathbf{g}||_2 \leq C$. We define the per-coordinate SNR in the preconditioned space, which is directly comparable to non-preconditioned DP-SGD:

$$\mathrm{SNR}_i := \frac{(\mathbb{E}[u_i])^2}{\mathrm{Var}(u_i)} \approx \frac{g_i^2/(\sigma^2 C^2)}{1} = \frac{g_i^2}{\sigma^2 C^2}. \tag{8}$$

Thus the SNR in DP-Adam matches that of DP-SGD. Therefore, DP turns each preconditioned coordinate into unit-variance noise with very small mean. A fixed-precision quantizer on $u_i$ then adds zero-mean noise whose variance does not depend on this small mean, so the signal-to-quantization-noise ratio behaves like $g_i^2/(\sigma^2 C^2)$ and deteriorates as privacy noise increases.

## 5 DPQUANT: OUR PROPOSED SOLUTION

### 5.1 PART I: PROBABILISTIC LAYER SAMPLING

Each time we perform randomized and unbiased quantization on a layer, we introduce additional variance to its gradient updates. While this added variance might be acceptable in standard training,

it results in a significant performance degradation under DP-SGD, where gradient stability is already challenged by injected noise.

Suppose a layer is quantized with probability $p$, we let $\mathbf{g}_{\text{fp}}$ denote its full precision gradients and $\mathbf{g}_{\text{quant}}$ to be its gradients computed under quantization. By Section 4, quantization incurs additional variance, hence $\text{Var}(\mathbf{g}_{\text{fp}}) \leq \text{Var}(\mathbf{g}_{\text{quant}})$. The *expected* gradient variance is:

$$\mathbb{E}\left(\text{Var}(\mathbf{g})\right) = (1-p)\,\text{Var}(\mathbf{g}_{\text{fp}}) + p\,\text{Var}(\mathbf{g}_{\text{quant}}) \leq \text{Var}(\mathbf{g}_{\text{quant}})$$

From this it follows that whenever $p < 1$ – that is, when only a subset of layers is quantized at each epoch – the average quantization-induced variance is strictly lower than in full quantization. Furthermore, by rotating which layers are quantized every epoch, no single layer repeatedly incurs the full quantization variance, and hence their expected variance remains smaller than $\text{Var}(\mathbf{g}_{\text{quant}})$.

## 5.2 PART II: LOSS-AWARE LAYER PRIORITIZATION

Not all layers contribute equally to model performance. Intuitively, we prefer to retain higher precision in layers that have a greater impact on the loss or accuracy. Given a constrained quantization budget, our goal is to prioritize quantization in lower-impact layers, thereby minimizing the overall loss in model quality.

We define a quantization policy $p$ to be the set of layers to compute under quantization. We define $R(p)$ as the expected *loss increase* incurred by applying quantization policy $p$ instead of full precision:

$$R(p) := \mathbb{E}_D\big[\mathcal{L}\big(M_p(D)\big) - \mathcal{L}\big(M_{\text{fp32}}(D)\big)\big].$$

Figure 2: DPQUANT system overview

Our goal is to find policies that minimally increase loss (or equivalently, a policy $p$ with small $R(p)$). We note that a key challenge in evaluating $R(p)$ for a given policy $p$ is that the expectation is taken over the full private dataset $D$. This makes direct computation both expensive and incompatible with tight privacy guarantees. To address this, we instead estimate $R(p)$ by subsampling $D$ and running a limited number of DP-SGD iterations under policy $p$ to obtain a proxy loss, then the same training iterations are done to obtain the baseline full-precision loss, and the difference is used as an empirical estimate of the loss impact.

Since this quantity is computed on the private training dataset $D$, any estimation of $R(\cdot)$ must be performed in a differentially private manner, and therefore consumes part of the overall privacy budget. Any computation using the private data incurs a privacy cost that must be accounted for to ensure that the privacy budget is not exceeded. We outline how DPQUANT privatizes and accounts for loss measurement in Section 5.4.

## 5.3 DYNAMIC LAYER SELECTION FOR QUANTIZED DP TRAINING

Building on the insights from the previous sections, we design a dynamic layer selection strategy for quantized DP training that combines: (i) probabilistic sampling of quantized layers to reduce variance (Section 5.1), and (ii) loss-aware prioritization to preserve performance by avoiding quantization of high-impact layers (Section 5.2).

Let $R(l_i)$ denote the estimated quantization loss impact of layer $i$. We define the probability of selecting layer $i$ for quantization as:

$$\pi_i := \frac{\exp(-\beta R(l_i))}{\sum_{j=1}^{n} \exp(-\beta R(l_j))}, \qquad \text{for } i = 1, \ldots, n,$$

where $\beta > 0$ is a scaling parameter that controls how strongly we prioritize low-impact layers.

As outlined in Figure 2, to quantize $k$ out of $n$ layers at each epoch, we sample a subset *without replacement* according to the distribution $\{\pi_i\}$. This allows us to adaptively choose the least sensitive

layers for quantization, while still randomly rotating layers with similar loss impact to minimize variance over time. This selection procedure is detailed in Appendix A.15. DPQUANT provides a set of tunable parameters that govern the frequency of the analysis, as well as other privacy parameters.

## 5.4 PRIVACY ACCOUNTING

Our method begins by measuring loss differences on each user's private dataset. Specifically, we compute $\mathcal{L}(M(D))$ which requires inspecting raw data and inherently risks exposing sensitive information if released directly. Without these privacy-preserving measures, simply publishing the loss-difference measurements compromises the privacy guarantee DP-SGD provides.

**Definition 3** (Sampled Gaussian Mechanism (SGM), (Mironov et al., 2019)). *Let $f$ be a function that maps subsets of a dataset $S$ to $\mathbb{R}^d$. The Sampled Gaussian Mechanism, denoted $\mathrm{SG}_{q,\sigma}$, is defined with sampling rate $0 < q \le 1$ and noise parameter $\sigma > 0$ as:*

$$\mathrm{SG}_{q,\sigma}(S) := f\left(\{x \in S : x \text{ is independently sampled with probability } q\}\right) + \mathcal{N}\left(0, \sigma^2 \mathbb{I}^d\right),$$

*where each element in $S$ is independently included with probability $q$, and $\mathcal{N}(0, \sigma^2 \mathbb{I}^d)$ denotes $d$-dimensional isotropic Gaussian noise with variance $\sigma^2$ per coordinate.*

To protect privacy, we frame this loss computation as a Sampled Gaussian Mechanism (SGM): we draw a random subsample of $D$, clip the resulting loss value to bound sensitivity, and then add Gaussian noise of scale $\sigma$. These operations correspond to step 3 of Algorithm 1.

---

**Algorithm 1** COMPUTELOSSIMPACT

1: **Input:** $P$ (policies), $B$ (batches), $R$ (iterations), $\alpha$ (decay), $\mathcal{C}$ (norm), $\sigma$ (noise)
2: Let $p_0$ be the baseline policy (no quantization)
3: Initialize a map for average losses, $\bar{\ell}$
4: **for** each $p \in P \cup \{p_0\}$ **do**          ▷ ❶ Compute avg. loss for baseline and all policies
5:     total_loss $\leftarrow 0$
6:     **for** $i = 1$ to $R$ **do**
7:         RESTOREMODEL()
8:         **for** each $(x, y) \in B$ **do**
9:             With policy $p$, run DPSGD-UPDATE($M, \mathrm{loss}(M(x), y)$)
10:        **end for**
11:        total_loss $\leftarrow$ total_loss $+ \frac{1}{|B|} \sum_{(x,y) \in B} \mathrm{loss}(M(x), y)$
12:    **end for**
13:    $\bar{\ell}[p] \leftarrow$ total_loss$/R$
14: **end for**
15: $R[p] \leftarrow \bar{\ell}[p] - \bar{\ell}[p_0]$ for all $p \in P$          ▷ ❷ Compute loss differences from baseline
16: $\mathbf{R} \leftarrow [R[p_1], \dots, R[p_k]]$
17: $\hat{\mathbf{R}} \leftarrow \mathbf{R} \cdot \min\left(1, \frac{\mathcal{C}}{||\mathbf{R}||_2}\right) + \mathcal{N}(0, \sigma^2 \mathcal{C}^2 \mathbb{1})$          ▷ ❸ Privatize differences
18: UPDATEPRIVACY( rate $= |B|/|D|$, steps $= 1$, noise_scale $= \sigma$)
19: **for** each $p \in P$ **do**
20:     $L[p] \leftarrow (1 - \alpha) \cdot L[p] + \alpha \cdot \hat{\mathbf{R}}[p]$          ▷ ❹ Update Policy EMA
21: **end for**
22: **return** $L$

---

**Proposition 2.** *Algorithm 1 is a Sampled Gaussian Mechanism (SGM) with sample rate $q = |B|/|D|$ and noise scale $\sigma = \sigma_{measure}$. Proof: See Appendix A.11.*

To account for the privacy cost we incur by performing the analysis in Algorithm 1, we rely on Opacus's privacy accountants (Yousefpour et al., 2022). This is for two reasons: First, these accountants measure the cumulative privacy loss of SGMs (Makni et al., 2025), where by Prop. 2 we can reuse its implementation. Second, by leveraging the advanced composition theorem (Abadi et al., 2016), we obtain a much tighter upper bound on the total privacy expenditure incurred by both the DP-SGD training process and any subsequent analyses performed under the same privacy budget. We explain this in more detail in Appendix A.14.

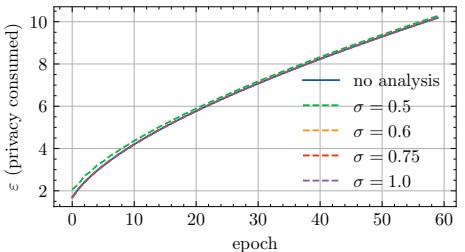 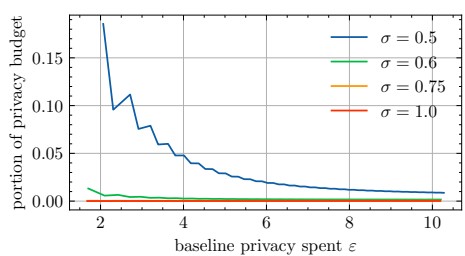

(a) privacy consumption of analysis + training

(b) Fraction of privacy spent on analysis

Figure 3: Privacy cost of analysis for ResNet18/GTSRB; performing analysis every 2 epochs

In Figure 3, we report the cumulative privacy loss from both the analysis and training components across various configurations. Our results empirically demonstrate that the privacy cost of analysis is negligible compared to training, and does not meaningfully affect the quality of the resulting model.

## 6 EVALUATION

**Models and Datasets.** We evaluate our approach on commonly used neural networks for differentially private training (Jagielski et al., 2020; De et al., 2022): ResNet18 (He et al., 2015), ResNet50 and DenseNet121 (Huang et al., 2018) from the `torchvision` (maintainers and contributors, 2016) library. We also test BERT (Devlin et al., 2019), trained with DP-AdamW. We additionally evaluate with DP-Adam in Appendix A.5. These models are trained on the Extended MNIST (Cohen et al., 2017), German Traffic Sign Recognition Benchmark (GTSRB) (Stallkamp et al., 2011), CIFAR-10 (Krizhevsky, 2009) and SNLI (Bowman et al., 2015) datasets.

**Implementation.** DPQUANT is implemented on top of Opacus (Yousefpour et al., 2022), a DP training framework which provides Poisson sampling, gradient clipping, and noising. The DPQUANT parameters can be found in Appendix A.2.

**Low Precision Format.** For low precision computations, we used the LUQ-FP4 (Chmiel et al., 2024) format, the highest-performing 4-bit quantization format. LUQ-FP4 uses a 4-bit representation of floating point numbers, consisting of 1 sign and 3 exponent bits. In Appendix A.9, we evaluate DPQUANT on other low-precision formats including FP8 and 4-bit uniform quantization.

### 6.1 QUANTIZATION-QUALITY TRADE-OFF

Quantizing more layers proportionally increases the speed of training. However, it also increases the accuracy degradation in DP-SGD training. Thus, there is a *speed-accuracy* trade-off depending on the number of layers quantized. For a given number of quantized layers, the resulting model accuracy can significantly vary depending on which layers are quantized at any given epoch. DPQUANT aims to automatically identify the subset of layers for each epoch that provides the best accuracy, assuming a certain number of layers are quantized. We refer to the desired number of quantized layers as "computational budget" because it determines the speed and compute resources needed.

In Figure 4, we sample $\approx 50$ random subsets of layers to execute in fp4. We plot the empirical Pareto front using these sampled measurements, in addition to the resulting validation accuracy when using DPQUANT's scheduling technique for a given computational budget.

We make two observations. First, we note that randomly selecting the quantized layers can lead to significant loss in accuracy, as much as $40\%$. Second, DPQUANT generates scheduling configurations that provide validation accuracy close to the Pareto-front for all evaluated networks and datasets.

### 6.2 SENSITIVITY TO PRIVACY BUDGET

We compared our method to the baseline for two privacy budgets $\varepsilon = 4$ and $\varepsilon = 8$. In Table 1 we report the validation accuracy for different privacy budgets. For ResNet18/50, we obtain these values

---

[1]Batch size decreased to improve convergence under $\varepsilon = 4$.

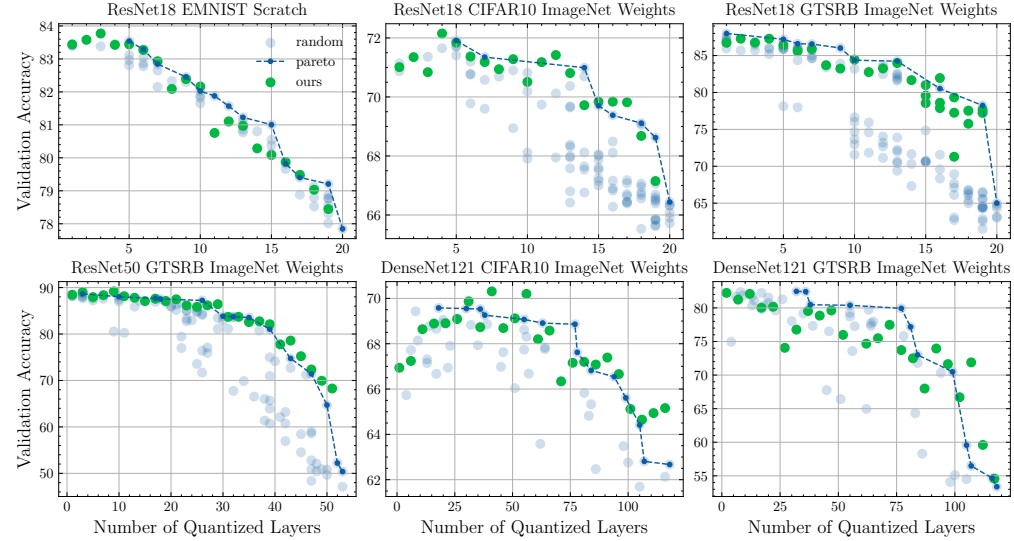

Figure 4: Comparing policies generated by DPQUANT to the speed-accuracy Pareto front

| Model | Dataset | Percent Quantized | $\varepsilon = 4$ | | | | $\varepsilon = 8$ | | | |
|---|---|---|---|---|---|---|---|---|---|---|
| | | | Baseline | $\varepsilon$ | Ours | $\varepsilon$ | Baseline | $\varepsilon$ | Ours | $\varepsilon$ |
| ResNet18 | EMNIST | 0.5 | 81.27 ± 1.29 | 3.14 | **82.16** | 3.04 | – | | | |
| | | 0.75 | 80.51 ± 0.37 | 3.01 | 80.09 | 3.04 | – | | | |
| | | 0.9 | 78.82 ± 0.30 | 3.01 | **79.03** | 3.04 | – | | | |
| | GTSRB | 0.5 | 42.34 ± 5.53 | 4.01 | **49.09** | 3.99 | 69.06 ± 5.63 | 8.01 | **76.75** | 7.99 |
| | | 0.75 | 39.98 ± 3.99 | 4.01 | **42.62** | 3.96 | 63.62 ± 5.59 | 8.01 | **70.07** | 7.99 |
| | | 0.9 | 37.94 ± 2.23 | 4.01 | **39.48** | 3.99 | 57.49 ± 4.46 | 8.01 | **67.67** | 7.99 |
| | CIFAR-10 | 0.5 | 64.37 ± 1.42 | 4.06 | **65.39** | 3.94 | 69.26 ± 1.46 | 7.12 | **70.51** | 7.17 |
| | | 0.75 | 62.17 ± 0.61 | 4.06 | **63.57** | 3.94 | 67.80 ± 0.81 | 7.12 | **69.84** | 7.17 |
| | | 0.9 | 61.09 ± 1.66 | 4.06 | 61.22 | 3.94 | 67.21 ± 1.24 | 7.12 | **68.68** | 7.17 |
| ResNet50 | GTSRB | 0.5 | 38.76 ± 8.16 | 4.01 | **42.11** | 3.99 | 75.99 ± 7.33 | 8.01 | **80.23** | 7.99 |
| | | 0.75 | 29.48 ± 4.72 | 4.01 | **33.67** | 3.99 | 58.13 ± 8.50 | 8.01 | **69.03** | 7.99 |
| | | 0.9 | 24.78 ± 2.67 | 4.01 | **29.00** | 3.99 | 47.40 ± 7.23 | 8.01 | **59.87** | 7.99 |
| DenseNet121 | GTSRB[1] | 0.5 | 54.10 ± 5.58 | 4.06 | 55.38 | 3.97 | 65.47 ± 5.42 | 8.01 | **71.05** | 7.93 |
| | | 0.75 | 44.60 ± 5.06 | 4.06 | **47.36** | 3.97 | 56.14 ± 7.57 | 8.01 | **63.30** | 7.93 |
| | | 0.9 | 40.52 ± 2.83 | 4.06 | **44.15** | 3.97 | 51.06 ± 5.41 | 8.01 | **52.60** | 7.93 |
| | CIFAR-10[1] | 0.5 | 59.22 ± 1.15 | 4.03 | **61.08** | 3.97 | 67.96 ± 0.93 | 7.12 | **68.96** | 7.28 |
| | | 0.75 | 56.43 ± 1.72 | 4.03 | **60.31** | 3.97 | 64.81 ± 1.71 | 7.12 | **66.48** | 7.28 |
| | | 0.9 | 55.18 ± 1.38 | 4.03 | **58.89** | 3.97 | 63.03 ± 1.69 | 7.12 | **65.13** | 7.28 |
| BERT | SNLI | 0.5 | | | | | 62.54 ± 4.54 | 7.48 | **67.80** | 7.48 |
| | | 0.75 | | | | | 52.04 ± 3.95 | 7.48 | **63.61** | 7.48 |

Table 1: Model quality across datasets and privacy levels.

by truncating the training at the respective privacy budgets (i.e. without additional hyperparameter tuning), and select baseline data points with larger $\varepsilon$ than ours wherever possible.

In most cases, DPQUANT outperforms the baseline performance by at least 1 standard deviation whilst not exceeding the privacy budget. In particular, despite the privacy cost of analysis being more dominant during the $\varepsilon = 4$ case, DPQUANT produces near-optimal quantization schedules, demonstrating its robustness with respect to $\varepsilon$.

We have also evaluated DPQUANT on extremely small privacy budgets (e.g. $\varepsilon = 1$). In Appendix A.3 we show that DPQUANT still demonstrates the same benefits under this setting.

### 6.3 ABLATION STUDY

To better understand the contributions of the two approaches, we compared our approach (probabilistic layer sampling + loss-aware layer prioritization) with probabilistic layer sampling (PLS) alone. In Figure 5, we observe that PLS consistently performs better than the baseline where the quantized

layers are selected statically. However, there is still a large gap between PLS and the *best-performing* layer selections, suggesting that some crucial layers are consistently being subjected to quantization which significantly degrades the quality of trained models.

When PLS is combined with loss-aware layer prioritization, the layers crucial to model training are left in full precision, even when most of the layers are quantized. The benefits of prioritization begins to surface as the proportion of quantized layers increase, as the critical layers have a larger probability of being quantized in the randomized baselines. Furthermore, we observe that the best training outcomes are achieved by combining both approaches. We include more details in Appendix A.7.

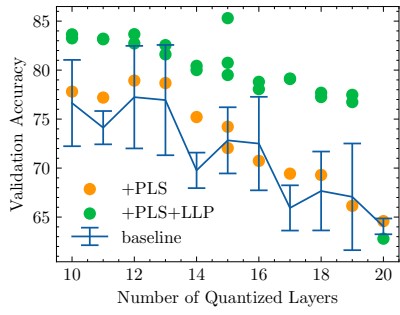

Figure 5: Ablation study, PLS: probabilistic layer selection, LLP: loss-aware layer prioritization

Figure 6: Theoretical speedups for DPQUANT assuming 90% of the layers are quantized.

## 6.4 THEORETICAL SPEEDUP

As hardware with support for FP4 MatMuls and Conv2D (e.g., NVIDIA Blackwell) are not yet widely available, we are unable to evaluate the speed benefits of quantization with DPQUANT. Instead, we use estimates from prior work, along with performance statistics published by NVIDIA (NVIDIA Corporation, 2024) to estimate speedups. Based on profiling on existing hardware, we estimate that FP4 can provide a $4\times$ speedup over the FP16 baseline. Separately, prior works (Sun et al., 2020; Choi et al., 2018; Abdolrashidi et al., 2021) report a $4 - 7.3\times$ speedup when using FP4 on supported hardware. To remain conservative, we use the lower bound ($4\times$) in our estimates. We assume matrix multiplications, convolutions, and element-wise operations can be accelerated $4\times$, and characterize the total runtime as a linear compute cost model:

$$T_{\text{ours}} = T_{\text{analysis}} + (1 - p + p/4)(T_{\text{train baseline}} - T_{\text{overhead}}) + T_{\text{overhead}}$$

where $T_{\text{analysis}}$ is the time taken by algorithm 1, and $T_{\text{overhead}}$ captures the time taken by operations that do not have performance benefits from low precision (details in appendix A.13).

We show our speedups in Figure 6. Quantized training with DPQUANT is $1.75\times$ to $2.21\times$ faster than the fp16 baseline. In particular, the loss-aware prioritization mechanism in DPQUANT incurs minimal runtime overhead, which is crucial to preserve the performance gains of fp4 computation.

## 7 CONCLUSION

In this paper, we introduce DPQUANT, a mechanism for efficient quantized DP training. We observe that existing quantized training techniques can significantly degrade the accuracy of models trained with DP-SGD and DP-Adam, and provide a detailed analysis demonstrating the amplified quantization error. To address this challenge, DPQUANT employs techniques to dynamically select layers to quantize such that the impact of quantization on model accuracy is minimized. DPQUANT itself is a differentially private mechanism that incurs only small privacy cost. We empirically demonstrate that DPQUANT achieves near-optimal compute-to-accuracy tradeoffs during quantized training, generalizes to different models, datasets and privacy budgets, and can provide up to $2.21\times$ speedup while minimally impacting accuracy. DPQUANT enables efficient and practical differentially-private training for both centralized and distributed training deployments.

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

## A  APPENDIX / SUPPLEMENTAL MATERIAL

### A.1  EFFECT OF BATCH SIZE

Our analysis of quantization error in DP-SGD is independent of batch size. While larger batches reduce stochastic gradient variance, our argument hinges on the magnitude of the final noised gradient, which remains large regardless of batch size.

1. In DP-SGD, the added noise's scale is proportional to the per-example clipping constant, $C$, not the batch size.

2. This noise dominates the averaged gradient signal, causing the raw gradients in subsequent steps to have significantly larger norms than in non-DP training.

3. As shown in Proposition 1, quantization variance is proportional to the square of the gradient's norm ($\mathrm{Var}(q(x)) = \Theta(||x||_\infty^2)$). Therefore, the larger gradients in DP-SGD lead to much higher quantization variance, which destabilizes training and degrades accuracy.

To demonstrate this empirically, we ran the same training job with batch sizes ranging from 1024 to 8192 and measured the numerical range of the weight gradients, similar to that in figure 1c. Across the batch sizes, there is negligible difference in the gradient ranges, which confirms our hypothesis.

Table 2: Weight gradient norm range across various batch sizes, showing the negligible impact of batch size on the final gradient magnitudes.

| Batch Size | Norm Range Mean | Norm Range Std |
|---|---|---|
| 1024 | 0.159 | 0.137 |
| 2048 | 0.161 | 0.127 |
| 4096 | 0.158 | 0.116 |
| 8192 | 0.156 | 0.119 |

### A.2  EVALUATION SETUP AND PARAMETERS

| Parameter | Default | Description |
|---|---|---|
| $n$ | 60 | number of epochs to train |
| $k$ | – | layers to execute in low precision. |
| $n_{\text{sample}}$ | 1 | test samples for loss measurement. |
| $n_{\text{interval}}$ | 2 | epochs to train before the next measurement. |
| $R$ | 2 | repetitions during measurement. |
| $\sigma_{\text{measure}}$ | 0.5 | Noise scale used during loss-difference privatization. |
| $C_{\text{measure}}$ | 0.01 | Clipping norm used during loss-difference privatization. |

Table 3: Configurable Hyperparameters of DPQUANT

**Remark: Selecting DPQUANT parameters in practice.**  In our experiments, we have found repetitions = 2 and sampling frequency = 1 to be the most optimal. Adopting these recommended defaults, the user needs to pick:

1. one of $k$ (number of layers to quantize) and the analysis frequency

2. clipping norm used in loss sensitivity analysis

The process of determining clipping norm for analysis is similar to that of finding the clipping threshold C for normal DP-SGD training. We want to pick a value such that the differences between policies are expressed.

## A.3 EVALUATION UNDER EXTREME PRIVACY BUDGETS

As shown in Figure 3, the privacy consumption of DPQUANT's analysis accounts for a higher fraction of the total privacy near the beginning of training. We wish to evaluate DPQUANT under more strict privacy budgets.

In these cases, both the parameters of DP-SGD and DPQUANT need to be updated, namely the noise scale $\sigma$ and measurement noise scale $\sigma_{\text{measure}}$ both need to be increased. Table 4 below shows that DPQUANT achieves optimal accuracy even when $\varepsilon = 1$.

Table 4: Training accuracy of ResNet18 with GTSRB under the strict privacy budget ($\varepsilon = 1$)

| | Baseline | | DPQUANT | |
|---|---|---|---|---|
| Count | Accuracy (%) | $\varepsilon$ | Accuracy (%) | $\varepsilon$ |
| 50% | $44.14 \pm 4.61$ | 1.05 | 48.26 | 0.99 |
| 75% | $40.13 \pm 3.65$ | 1.05 | 43.14 | 0.99 |
| 90% | $35.20 \pm 1.12$ | 1.05 | 38.66 | 0.99 |

## A.4 TRAINING HYPERPARAMETERS

### A.4.1 IMAGE MODELS

While the learning rate might seem too high for regular SGD training, previous results Morsbach et al. (2024); Ponomareva et al. (2023) have shown that large learning rates are more beneficial for DP-SGD training.

Table 5: Experimental configurations (6 runs)

| | 1 | 2 | 3 | 4 | 5 | 6 |
|---|---|---|---|---|---|---|
| Model | ResNet18 | ResNet18 | ResNet18 | ResNet50 | DenseNet121 | DenseNet121 |
| Dataset | EMNIST | CIFAR10 | GTSRB | GTSRB | CIFAR10 | GTSRB |
| $\sigma$ | 1 | 1 | 1 | 1 | 1 | 1 |
| $\delta$ | $10^{-5}$ | $10^{-5}$ | $10^{-5}$ | $10^{-5}$ | $10^{-5}$ | $10^{-5}$ |
| Clipping norm | 1 | 1 | 1 | 1 | 1 | 1 |
| Batch size | 1024 | 1024 | 1024 | 1024 | 512 | 512 |
| Physical batch size | 128 | 128 | 128 | 128 | 128 | 128 |
| Weights | None | ImageNet | ImageNet | ImageNet | ImageNet | ImageNet |
| Optimizer | SGD | SGD | SGD | SGD | SGD | SGD |
| Learning rate (lr) | 0.5 | 0.5 | 0.5 | 0.5 | 0.5 | 0.5 |
| Epochs | 30 | 60 | 60 | 60 | 60 | 60 |

### A.4.2 LANGUAGE MODELS

We conducted a new NLP experiment using BERT for sequence classification on the Stanford Natural Language Inference (SNLI) corpus. In this task, the model classifies a pair of statements (e.g., "Children smiling and waving at camera" and "There are children present") as "entailment," "contradiction," or "neutral."

Due to the high number of parameters in BERT, we have followed the tutorial from Opacus and frozen 12 out of 13 BERT layers, and trained the last BERT layer and subsequent classification layers.

We have compared our method (DPQUANT) with a random static baseline (similar to section 6.1). We use the same training parameters, trained for a single epoch, and used $\varepsilon = 8$ as the total privacy budget.

In these experiments, DPQUANT outperforms the baseline in accuracy. DPQUANT consistently avoids quantizing the last few layers (including the trainable ones) without prior information about the importance and trainability of the layers.

## A.5 EVALUATION WITH DP-ADAM

We evaluate DPQUANT using DP-Adam as the optimizer. All hyperparameters match the DP-SGD configuration in Table 5, with the optimizer changed to Adam (learning rate 0.01, default $\beta_1 = 0.9$, $\beta_2 = 0.999$).

Table 6 shows that DPQUANT consistently outperforms the static baseline under DP-Adam while consuming a strictly lower privacy budget, with the largest gains at 75% and 90% quantization.

Table 6: DP-Adam: DPQUANT vs. static random baseline (ResNet18).

| | | Baseline | | DPQUANT | |
|---|---|---|---|---|---|
| **Dataset** | **Percent Quantized** | **Accuracy (%)** | $\varepsilon$ | **Accuracy (%)** | $\varepsilon$ |
| CIFAR-10 | 0.50 | $54.18 \pm 1.48$ | 5.05 | **55.03** | 4.97 |
| | 0.75 | $49.86 \pm 1.57$ | 5.05 | **52.40** | 4.97 |
| | 0.90 | $47.36 \pm 1.36$ | 5.05 | **51.18** | 4.97 |
| GTSRB | 0.50 | $72.74 \pm 6.04$ | 6.05 | **75.87** | 5.91 |
| | 0.75 | $58.14 \pm 7.64$ | 6.05 | **72.32** | 5.91 |
| | 0.90 | $50.88 \pm 5.11$ | 6.05 | **66.45** | 5.91 |

Additionally, the BERT experiment on SNLI (Table 1) uses DP-AdamW as the underlying optimizer.

## A.6 ACCURACY DEGRADATION FOR DP-SGD UNDER NAIVE QUANTIZATION

Prior works Sun et al. (2020); Chmiel et al. (2024); Micikevicius et al. (2022) have demonstrated minimal degradation during quantized fp4/8 training compared to the full precision counterpart. We tabulate their results below:

Table 7: Ultra-Low and LUQ vs. baseline accuracy

| Model | Baseline | Ultra-Low Sun et al. (2020) | LUQ Chmiel et al. (2024) |
|---|---|---|---|
| ResNet-18 | 69.7% | 68.27% (-1.43%) | 69.09% (-0.61%) |
| ResNet-50 | 76.5% | 74.01% (-2.49%) | 75.42% (-1.08%) |
| MobileNet-V2 | 71.9% | 68.85% (-3.05%) | 69.55% (-2.35%) |
| ResNext50 | 77.6% | N/A | 76.02% (-1.58%) |
| Transformer-base | 27.5 (BLEU) | 25.4 (-2.10) | 27.17 (-0.33) |

As demonstrated in the Figure 4, the performance degradation of DP-SGD under quantization is much larger.

Table 8: Validation accuracy for DP-SGD training: baseline vs. LUQ-FP4 (all layers quantized)

| Model | Dataset | Pretraining | Baseline | LUQ-FP4 | $\Delta$ |
|---|---|---|---|---|---|
| ResNet-18 | EMNIST | None | 83.4% | 77.8% | -5.6% |
| ResNet-18 | CIFAR-10 | ImageNet | 71.0% | 65.8% | -5.2% |
| ResNet-18 | GTSRB | ImageNet | 85.6% | 64.0% | -21.6% |
| ResNet-50 | GTSRB | ImageNet | 89.8% | 49.0% | -40.8% |
| DenseNet-121 | CIFAR-10 | ImageNet | 67.0% | 62.9% | -4.1% |
| DenseNet-121 | GTSRB | ImageNet | 82.0% | 53.0% | -29.0% |

## A.7 SENSITIVITY OF TEMPERATURE $\beta$

In our method, the temperature parameter $\beta$ provides a crucial mechanism to balance two complementary strategies:

- Deterministic Selection: This approach prioritizes elements based on their loss sensitivity, selecting those that are most impactful.
- Randomized Sampling: This approach introduces stochasticity, ensuring diversity and exploration in the selection process.

A low $\beta$ value favors randomized sampling, while a high $\beta$ value makes the selection process more deterministic and reliant on loss sensitivity. Below, we tabulate the training accuracy for different value of $\beta$, and observe that better training outcomes can be obtained by favoring loss-based layer selection while retaining some stochasticity. Namely, it performs strictly better than selection purely based on random layer sampling.

Table 9: Model performance across various counts and temperature ($\beta$) values

| Count | Temperature ($\beta$) | | | | | | | | |
|---|---|---|---|---|---|---|---|---|---|
| | 0.1 | 0.22 | 0.47 | 1.03 | 2.24 | 4.86 | 10.57 | 22.99 | 50.0 |
| 10 | 66.49 | 67.58 | 67.53 | 67.01 | 70.25 | 70.37 | **71.59** | 70.96 | **71.67** |
| 15 | 58.47 | 58.47 | 60.03 | 59.07 | 60.86 | **65.00** | 60.54 | **65.04** | 63.75 |
| 18 | 51.60 | 54.08 | 55.73 | 53.73 | 53.86 | 53.49 | **60.90** | 55.45 | 56.05 |

## A.8 EMA ABLATION

The exponential moving average (EMA) in step ❹ of Algorithm 1 smooths the per-layer importance scores across measurement iterations so that a single noisy batch does not drastically change the layer ranking. Table 10 shows the effect of removing EMA on ResNet18 trained on CIFAR-10 with DP-SGD. Across all quantization levels, EMA consistently improves accuracy, confirming that smoothing the noisy sensitivity estimates is beneficial.

Table 10: Effect of EMA on DPQUANT (ResNet18, CIFAR-10, DP-SGD).

| Percent Quantized | With EMA | Without EMA |
|---|---|---|
| 0.50 | 56.06 | 54.01 |
| 0.75 | 51.59 | 50.07 |
| 0.90 | 49.97 | 47.22 |

## A.9 EVALUATION ON OTHER QUANTIZERS

To assess the versatility of our method, DPQuant, we evaluated its performance with different numerical precisions and quantization schemes. We conducted two primary experiments: one using 8-bit floating-point (`fp8e5m2`) for training to test a different bitwidth, and another using a uniform 4-bit quantizer to test a different quantization strategy.

### A.9.1 FP8 QUANTIZATION

Our experiments with FP8 training show that quantized DP-SGD does not suffer a significant performance degradation. This minimal performance gap suggests that in higher-precision settings like FP8, the benefits of more complex techniques like layer subset selection may be less critical. We show the results in table 11.

Table 11: Performance comparison with FP8 training.

| Count | Base Acc(%) | Base $\epsilon$ | Our Acc(%) | Our $\epsilon$ |
|---|---|---|---|---|
| 50% | $67.56 \pm 0.47$ | 4.05 | 67.12 | 3.93 |
| 75% | $67.76 \pm 0.67$ | 4.05 | 67.65 | 3.93 |
| 90% | $67.38 \pm 0.59$ | 4.05 | 68.01 | 3.93 |

### A.9.2 UNIFORM 4-BIT QUANTIZATION

Next, we evaluated a more aggressive quantization scheme using a uniform FP4 quantizer. In this setup, the value range is discretized into $2^4 = 16$ levels via stochastic rounding. The results reveal a more substantial drop in accuracy for our method compared to the baseline. This outcome is consistent with our observations of the LUQ-FP4 quantizer discussed in Section 6.2, highlighting the inherent challenges of applying DP-SGD with very low-bitwidth uniform quantization. We show the results in table 12.

Table 12: Performance comparison with uniform FP4 quantization.

| Count | Base Acc(%) | Base $\epsilon$ | Our Acc(%) | Our $\epsilon$ |
|---|---|---|---|---|
| 50% | $63.56 \pm 0.89$ | 4.53 | 62.15 | 4.44 |
| 75% | $57.85 \pm 0.90$ | 4.53 | 59.09 | 4.44 |
| 90% | $55.82 \pm 0.80$ | 4.53 | 56.27 | 4.44 |

### A.10 PROOF OF PROPOSITION 1

**Proposition 1.** *Let $q : \mathbb{R}^n \to \mathbb{R}^n$ be an unbiased ($\mathbb{E}[q(\mathbf{x})] = \mathbf{x}$) and scale-invariant ($q(\lambda \mathbf{x}) = \lambda q(\mathbf{x})$) quantizer whose outputs lie on a fixed finite grid. If $\mathbf{x}$ is drawn from an absolutely continuous distribution, then*

$$\mathrm{Var}\big(q(\mathbf{x})\big) = \Theta\big(||\mathbf{x}||_\infty^2\big).$$

*Proof.* We begin by first showing the upper-bound: $\mathrm{Var}\left(q(\mathbf{x})\right) = \mathcal{O}\left(||\mathbf{x}||_\infty^2\right)$. We define $M = ||\mathbf{x}||_\infty$ and $\mathbf{v} = \mathbf{x}/M$, so $||\mathbf{v}||_\infty = 1$. By scale-invariance of $q$,

$$\mathrm{Var}\big(q(\mathbf{x})\big) = \mathrm{Var}\big(q(M\mathbf{v})\big) = \mathrm{Var}\big(M\,q(\mathbf{v})\big) = M^2\,\mathrm{Var}\big(q(\mathbf{v})\big).$$

Since $q(\mathbf{v}) \in [-1, 1]^n$, there exists a finite $C$ such that $\mathrm{Var}(q(\mathbf{v})) \leq C$, giving

$$\mathrm{Var}\big(q(\mathbf{x})\big) = M^2 \,\mathrm{Var}\left(q(\mathbf{v})\right) \leq C\,M^2 = C\,||\mathbf{x}||_\infty^2.$$

Next, we show the lower-bound: $\mathrm{Var}\left(q(\mathbf{x})\right) = \Omega\left(||\mathbf{x}||_\infty^2\right)$. On the compact set $\{\mathbf{v} : ||\mathbf{v}||_\infty = 1\}$, the continuous function $v \mapsto \mathrm{Var}(q(\mathbf{v}))$ attains a minimum $m \geq 0$. Because the finite quantizer grid has measure-zero, absolute continuity of $\mathbf{x}$ ensures the probability of $\mathbf{v}$ landing on the grid is 0, so $\mathrm{Var}(q(\mathbf{v})) > 0$ and hence $m > 0$. Therefore

$$m\,M^2 \leq \mathrm{Var}\big(q(\mathbf{x})\big) \leq C\,M^2,$$

From this we conclude $\mathrm{Var}(q(\mathbf{x})) = \Theta(||\mathbf{x}||_\infty^2)$, as desired. $\square$

### A.11 JUSTIFICATION OF PRIVACY GUARANTEES (THEOREM 2)

**Proposition 2.** *Algorithm 1 is a Sampled Gaussian Mechanism (SGM) with sample rate $q = |B|/|D|$ and noise scale $\sigma = \sigma_{measure}$.*

*Proof.* We first characterize Algorithm 1 as an analysis on the user's private dataset. The function accepts a subsampled batch of size $|B|$ from a dataset with $|D|$ samples.

Using this batch of user data, as well as some non-private sources of data such as the model weights, we compute the loss differences which is vectorized in $\mathbf{R} \in \mathbb{R}^p$, where $p$ is the number of available quantization policies.

In step ❸ of Algorithm 1, we clip the vector $\mathbf{R}$ to norm $\mathcal{C}$, to which independent Gaussian noise proportional to $\sigma^2 \mathcal{C}^2$ is added to obtain $\hat{\mathbf{R}}$. This is equivalent to adding noise proportional to $\sigma^2$ when the sensitivity of $\hat{\mathbf{R}}$ is 1 through a scaling argument.

Algorithm 1 ceases to access private data in $B$ after the computation of $\hat{\mathbf{R}}$, which results in step ❹ (updating the policy EMA) being post-processing Dwork and Roth (2014) which does not impact the privacy consumption.

Furthermore, the privacy accounting step makes use of Opacus' Yousefpour et al. (2022) privacy accountant, which assumes[2] the noise scale $\sigma$ is proportional to the clipping constant (i.e. equivalent to adding a noise proportional to $\sigma^2\mathcal{C}^2$. □

### A.12 LOW PRECISION SIMULATION SETUP

As FP4 hardware support is forthcoming, we employ the following simulation setup to emulate the effect of training under FP4. Notably, we quantize both inputs to the conv2d forward, wgrad, and dgrad operators as well as its output.

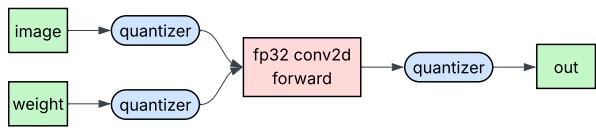

Figure 7: Quantization simulation setup

### A.13 THEORETICAL SPEEDUP CALCULATION

Due to the unavailability of accelerators and reliable software support for FP4, we instead rely on a performance model to estimate the theoretical throughputs of FP4 computation.

We first decompose the DP-SGD training computation into the following operations, listed in table 13.

We performed profiling on the models (on their respective datasets) stated in the paper, and we plot the runtime decomposition in Figure 8. Using this data, we can compute the amount of "overhead" (i.e. the time spent on operators which will not benefit from lower precision) for each model/dataset. This is tabulated in Table 14.

Table 13: Decomposition of DP-SGD training

| Computation Stage | Description | Benefits from FP4 |
|---|---|---|
| *Total Forward* | Time spent on the forward pass through the model, where input data is processed layer by layer to produce the output. | ✓ |
| *Total Backward* | Time for backpropagation, where gradients are calculated for model parameter updates. | ✓ |
| *Optimizer Clip* | Time for clipping gradients to a predefined threshold to ensure stability and prevent large updates during training. | ✓ |
| *Optimizer Noise* | Time for adding random noise to the gradients to ensure differential privacy by masking individual data point contributions. | |
| *Optimizer Scale* | Time for scaling the gradients after clipping to adjust the magnitude of the updates. | ✓ |
| *Other Optimizer* | Time spent on other optimizer-related operations, such as learning rate management. | |
| *Other Time* | Time for all other operations during the training iteration, including data loading, synchronization, and auxiliary tasks. | |

---

[2]This assumption is stated in `https://github.com/pytorch/opacus/blob/main/opacus/accountants/analysis/r`

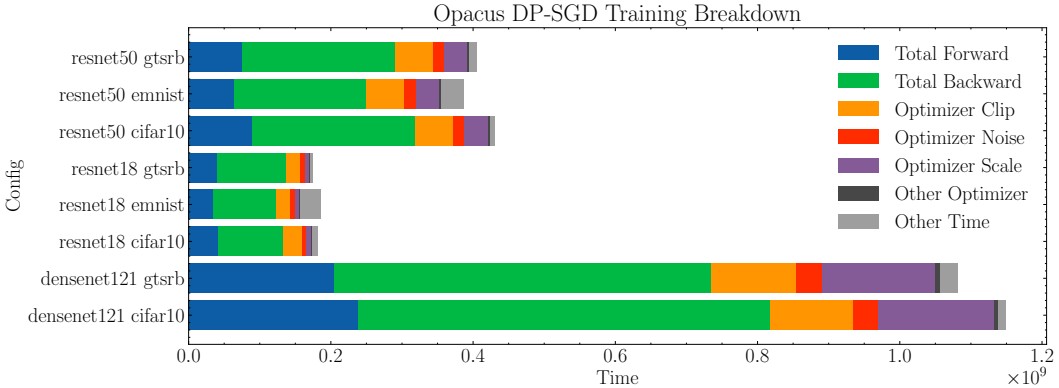

Figure 8: Runtime decomposition of DP-SGD training

Table 14: Breakdown of total time, good ops, bad ops, and overhead percentage for different model configurations.

| Config | Total Time | Ops with Speedup | Overhead Ops | Overhead % |
|---|---|---|---|---|
| DenseNet121 CIFAR10 | $1.15 \times 10^9$ | $1.10 \times 10^9$ | $5.23 \times 10^7$ | 4.55 |
| DenseNet121 GTSRB | $1.08 \times 10^9$ | $1.01 \times 10^9$ | $6.74 \times 10^7$ | 6.23 |
| ResNet18 CIFAR10 | $1.82 \times 10^8$ | $1.66 \times 10^8$ | $1.68 \times 10^7$ | 9.20 |
| ResNet18 EMNIST | $1.86 \times 10^8$ | $1.49 \times 10^8$ | $3.68 \times 10^7$ | 19.81 |
| ResNet18 GTSRB | $1.74 \times 10^8$ | $1.63 \times 10^8$ | $1.04 \times 10^7$ | 5.99 |
| ResNet50 CIFAR10 | $4.31 \times 10^8$ | $4.05 \times 10^8$ | $2.55 \times 10^7$ | 5.92 |
| ResNet50 EMNIST | $3.88 \times 10^8$ | $3.36 \times 10^8$ | $5.13 \times 10^7$ | 13.22 |
| ResNet50 GTSRB | $4.05 \times 10^8$ | $3.76 \times 10^8$ | $2.87 \times 10^7$ | 7.10 |

### A.14 OPACUS PRIVACY ACCOUNTING

Opacus maintains a tuple of the form (sample rate, noise scale, number of steps), which is incrementally updated during training. At any point, we can query the current privacy cost in terms of $(\varepsilon, \delta)$ by specifying a target $\delta$ and using either the `rdp` or `prv` accountant. This mechanism enables flexible and precise tracking of privacy usage, allowing us to assess how much additional privacy is consumed by our analysis relative to standard training.

### A.15 SAMPLING OF QUANTIZED LAYERS

We outline the algorithm DPQUANT uses to select layers to compute under quantization in Algorithm 2.

---

**Algorithm 2** SELECTTARGETS

---

1: **Input:** $L$ (EMA scores), $P$ (set of policies), $s$ (temperature), $m$ (number to sample), `layers` (set of layers to quantize under policy $p$)
2: $v \leftarrow [L[p]$ for $p \in P]$
3: $v \leftarrow (v - \min(v))/(\max(v) - \min(v))$                    ▷ Normalize
4: $\pi \leftarrow$ `softmax`$(-s \cdot v)$
5: $Q \leftarrow$ Multinomial$(\pi, m,$ without replacement)              ▷ Sample $m$ policies
6: $S \leftarrow \emptyset$
7: **for** each $p \in Q$ **do**
8:     $S \leftarrow S \cup$ `layers`$[p]$
9: **end for**
10: **return** $S$

---

### A.16 Remark: Applicability to Non-DP Training

The mechanisms in Sections 5.1 and 5.2 are general and could in principle apply to non-DP settings. However, prior work such as LUQ-FP4 (Chmiel et al., 2024) shows that FP4 non-DP training suffers negligible degradation, so DPQuant's improvements would be marginal there. Our main novelty and motivation are DP-specific: we identify and analyze a setting where non-DP FP4 training shows negligible degradation, but DP + FP4 suffers substantial accuracy loss (Figure 1a), attributing this to the interaction between $\ell_2$-calibrated DP noise and $\ell_\infty$-driven quantization. Furthermore, our loss-sensitivity measures and selection rules are designed to be compatible with DP accounting and composition, relying only on quantities that are already DP-released or incorporable into the privacy accountant, which is not a concern in standard non-DP quantization work.

### A.17 Remark: Vulnerability to Floating Point Attacks

Differential privacy implementations must carefully consider the vulnerabilities highlighted by Mironov (2012). Mironov identified that the floating-point implementation of noise sampling for mechanisms such as Laplacian or Gaussian introduces a "porous" distribution that lacks translation invariance. This issue is prevalent in both fp64 and fp32 arithmetic.

To ensure robustness against this vulnerability, our method has been meticulously designed. The critical step of noise addition in our framework occurs under standard conditions, prior to the application of our novel quantization technique. The process is as follows:

1. Gradients are maintained in full fp32 precision.
2. Noise is sampled and added to these fp32 gradients, also in fp32 precision. Only after the noisy gradient is computed is it quantized for use in the forward/backward pass of select layers.
3. Thus, the noise injection process maintains a vulnerability profile identical to that of standard DP-SGD implemented in fp32. The use of lower-precision representations for computation does not alter or exacerbate the known properties of the initial noise addition.

Additionally, our method is fully compatible with established defenses against this vulnerability. The 'snapping mechanism' proposed by Mironov, a post-processing step applied directly to the noisy output, would be applied to the full-precision fp32 gradients immediately after noise addition and before quantization in our pipeline.

