# OpenReview forum: "DPQuant: Efficient and Private Model Training via Dynamic Quantization Scheduling"
_ICLR.cc/2026/Conference — ICLR 2026 Poster_

### Official Review · Reviewer_ieCQ · 2025-10-28

**Soundness:** 3
**Presentation:** 1
**Contribution:** 2
**Rating:** 4
**Confidence:** 3

**Summary:**

This paper investigates dynamic quantization for differentially private SGD. The authors propose a specific method that converts specific layers into a low-precision format and design a selection criterion based on the difference between the quantized and non-quantized layers. They empirically validated the strength of their dynamic quantization approach across various settings, including different configurations and datasets.

**Strengths:**

•  The paper investigates the quantization of differentially private deep learning.

•  The authors provide some basic analysis of why DP training interacts poorly with quantization.

•  Based on their observations, the authors’ new training method enables faster training compared to FP32 training.

**Weaknesses:**

Please refer to the Questions section.

**Questions:**

•	In the related work section, the authors mention that all existing methods are orthogonal to this work. However, for readers who are not familiar with quantization, it is hard to understand the relationship between these methods and the authors' work. The authors need to improve their related work section. Maybe including more on DP network pruning or layer selection, such as [1].

[1] Differential Privacy Meets Neural Network Pruning, arXiv:2303.04612

•	The reviewer is not convinced why the authors shift their perspective from the L2 norm (for noise addition) to the L-inf norm (for analysis). Can the authors clarify this reason or explain why the L2 norm is not sufficient for their analysis? As far as I understand, Figure 1 does not support the reason for the L-inf analysis, and the empirical part uses the L2 norm.

•	Does the quantization method meet the properties in Proposition 1, such as the unbiased and scale-invariant properties?

•	If a low-precision format induces the DP accuracy drop, can using higher precision mitigate the accuracy drop in DP training?

•	When quantizing, is there any difference between floating-point and fixed-point quantization in terms of differential privacy?

•	The authors should carefully check the paper for readability. For example, the GTSRB dataset is not explained or cited when it is first used. And the indicator function in Eq. 1 should be clarified.

---

> ### Author Response · Authors · 2025-11-24
>
> Thank you for your valuable and detailed feedback on our paper. Below, we address each of your concerns in detail.
>
> ## Relationship with Pruning.
>
> We thank the reviewer for this suggestion and will add the suggested citations and clarification to the related work section.
>
> Conceptually, these methods target a different optimization objective than DPQuant. Pruning and layer-selection approaches primarily aim to improve the model–privacy trade-off: by reducing the effective dimensionality of the parameter space, they can use the privacy budget more frugally for a given utility target. However, even after pruning or layer selection, one still performs (almost) a full forward pass and a substantial backward pass, so the cost of executing the dense computation graph on hardware is not reduced nearly as much. In contrast, DPQuant assumes a fixed dense architecture and DP mechanism, and instead optimizes how much of the training computation can be executed in low precision to improve runtime and hardware efficiency under DP.
>
> ## $\ell_2$ vs. $\ell_\infty$ norm analysis.
>
> The two norms arise from two distinct sources. The DP mechanism follows standard DP-SGD: per-example gradients are clipped in $\ell_2$ to a bound C, and Gaussian noise is added with variance calibrated to C, so the privacy guarantee and accounting are naturally expressed in $\ell_2$.
>
> By contrast, the quantizer itself cares about the $\ell_\infty$ norm: to avoid overflow and fit within the FP4 (or INT4) dynamic range, each tensor or block is normalized by its maximum absolute entry, so the quantization step size - and thus the rounding error - are controlled by $||g||_\infty$.
>
> Thank you for raising this point - we will clarify this further in the paper.
>
> ## Unbiasedness and Scale Invariance.
>
> Yes, the quantizer we use satisfies the assumptions in Proposition 1. In practice, we first rescale each tensor by a shared scale factor and then apply stochastic rounding onto the FP4 grid. The shared scaling makes the quantizer effectively scale-invariant (multiplying the tensor by a constant just rescales the factor), and stochastic rounding ensures that it is unbiased in expectation.
>
> These assumptions are also quite standard in the low-precision training literature: formats such as MXFP4 and NVFP4 similarly rely on shared scaling for stability, while LUQ-FP4 explicitly uses unbiased stochastic rounding. Thus, Proposition 1 is not tailored to a special-case quantizer but covers the common class of practical 4-bit training schemes, including ours.
>
> ## Does higher precision mitigate accuracy drop?
>
> Yes. In the appendix, our FP8 experiments show that DP training with FP8 consistently achieves accuracy much closer to FP32 than FP4 at the same privacy budget and number of steps, confirming that moving to a higher precision mitigates the DP accuracy drop induced by very low-precision formats. However, current hardware is able to achieve much better performance with FP4 due to the lower arithmetic intensity.
>
> ## Floating vs. fixed point quantization.
>
> From a DP standpoint, FP4 and INT4 are both post-processing of the same clipped-and-noised gradients, so the formal privacy guarantee is unchanged. Our INT4 (uniform) experiments in the appendix show that DP training under INT4 also experiences noticeable degradation relative to FP32, and that DPQuant similarly narrows this gap compared to a static INT4 baseline.
>
> ## Paper readability.
>
> Thank you for the suggestion! We will clarify these issues and improve the general readability of the paper.
>
> Once again, we thank you for your detailed and helpful review. We hope that our responses and planned revisions have fully addressed your concerns. In light of these clarifications, we would be grateful if you would consider increasing your score.

---

### Official Review · Reviewer_ixjt · 2025-11-01

**Soundness:** 2
**Presentation:** 3
**Contribution:** 2
**Rating:** 4
**Confidence:** 3

**Summary:**

This paper focuses on differential privacy (DP) and the impact of quantization variance. The authors hypothesize that quantization variance leads to disproportionately large accuracy degradation in DP settings. To address this issue, they propose DPQUANT, a dynamic quantization framework that adaptively selects a changing subset of layers to quantize at each epoch. The main procedure consists of (i) probabilistic sampling of layers and (ii) loss-aware layer prioritization.

**Strengths:**

The mathematical analysis is clearly presented, and the theoretical treatment of differential privacy in the proposed algorithm, such as the post-processing of the update exponential moving average (EMA), appears sound. The empirical settings are well constructed, and the figures and tables are clear and easy to interpret. Compared with vanilla DP-SGD, the proposed method demonstrates meaningful performance improvements.

**Weaknesses:**

However, the related work and comparison baselines raise significant concerns. Quantization is a widely used technique across various applications, and it is natural to consider its potential benefits within differential privacy. Indeed, several prior works have explored the interplay between quantization and DP, including:
- Kang, Tianqu, et al. "The effect of quantization in federated learning: a rényi differential privacy perspective." 2024 IEEE International Mediterranean Conference on Communications and Networking (MeditCom). IEEE, 2024.
- Youn, Yeojoon, et al. "Randomized quantization is all you need for differential privacy in federated learning." arXiv preprint arXiv:2306.11913 (2023).
- Kim, Muah, Onur Günlü, and Rafael F. Schaefer. "Effects of quantization on federated learning with local differential privacy." GLOBECOM 2022-2022 IEEE Global Communications Conference. IEEE, 2022.
- Wang, Yongqiang, and Tamer Başar. "Quantization enabled privacy protection in decentralized stochastic optimization." IEEE Transactions on Automatic Control 68.7 (2022): 4038-4052.
-Xiong, Sijie, Anand D. Sarwate, and Narayan B. Mandayam. "Randomized requantization with local differential privacy." 2016 IEEE International Conference on Acoustics, Speech and Signal Processing (ICASSP). IEEE, 2016.
While some of these works focus on federated learning, the authors should still acknowledge them and discuss how their method relates to or differs from these approaches.

Beyond these, it is also important to consider recent advances in DP optimization methods beyond DP-SGD. Since DP-SGD is known to suffer from excessive noise and slow convergence, relying solely on it as a baseline is insufficient to demonstrate the full benefit of the proposed approach. Several improved variants could be relevant comparisons:
- Wang, Zihao, et al. "{DPAdapter}: Improving Differentially Private Deep Learning through Noise Tolerance Pre-training." 33rd USENIX Security Symposium (USENIX Security 24). 2024.
- Wei, Jianxin, et al. "Dpis: An enhanced mechanism for differentially private sgd with importance sampling." Proceedings of the 2022 ACM SIGSAC Conference on Computer and Communications Security. 2022.
- Park, Jinseong, et al. "Differentially private sharpness-aware training." International Conference on Machine Learning. PMLR, 2023.
- Denisov, Sergey, et al. "Improved differential privacy for sgd via optimal private linear operators on adaptive streams." Advances in Neural Information Processing Systems 35 (2022): 5910-5924.

In summary, while the proposed method is technically sound and the presentation is clear, the related work section and choice of baselines need substantial improvement to strengthen the novelty and empirical validity of this paper.

If these concerns are properly addressed, I would be pleased to raise my evaluation score.

**Questions:**

It is unclear whether the Exponential Moving Average (EMA) update plays an essential role in the proposed framework. While the paper mentions EMA as part of the post-processing step for maintaining differential privacy, there is no ablation study isolating its effect. It would strengthen the work to report performance both with and without EMA, or to justify why EMA is necessary for stability or privacy preservation in DPQUANT.
Furthermore, it would be helpful to discuss whether other optimization or averaging techniques (e.g., momentum-based methods, adaptive optimizers such as Adam or Adagrad, or other moving-average variants) could serve a similar role. Clarifying whether the proposed approach is tightly coupled with EMA or can generalize to other optimizers would provide deeper insight into the method’s robustness and applicability.

---

> ### Author Response · Authors · 2025-11-24
>
> Thank you for your valuable and detailed feedback on our paper. Below, we address each of your concerns in detail.
>
> ## Expanding related works section.
>
> We thank the reviewer for providing these citations and agree that our related-work section can more clearly situate our contribution. In the revised version, we will expand this section to explicitly discuss these lines of work and their relationship to ours.
>
> These papers, however, pursue goals that are largely orthogonal and complementary to those of DPQuant.
>
> The first group of citations propose either to **use quantization itself as the privacy mechanism** (Youn et al. 2023, Wang & Başar 2022) or use quantization as a way to compress gradients while providing DP guarantees (Kang et al. 2024, Kim et al. 2022) or both (Xiong et al. 2016) in a **federated/decentralized learning setup**.
>
> In these approaches, quantization is applied after gradients have been computed in full precision, so there are no downstream effects on subsequent forward/backward passes. By contrast, our goal is to speed up the training computation itself: we quantize intermediate activations and weights during training, so quantization error can propagate across layers and interact with DP noise, which necessitates the more detailed layer-wise variance analysis and scheduling procedure developed in this work.
>
> The second group seek to **change the DP learning algorithm / privacy mechanism itself** to get better accuracy for a fixed privacy budget ($\varepsilon, \delta$).
>
> Some lessons such as pretraining are already used in our work, as many of our image models and language models begin with pretrained weights using public datasets (such as ImageNet). We have shown that DPQuant is also robust to other optimizers such as Adam and AdamW, and we believe our contributions would be orthogonal to this group of prior works.
>
> In contrast, in our work we develop a hardware-efficient quantized training procedure that reduces quantization-induced variance and preserves accuracy under DP, especially on low-precision accelerators.
>
> ## Justification for EMA
>
> We thank the reviewer for this observation. EMA is a light-weight heuristic to smooth the per-layer importance scores across testing iterations so that a single noisy batch does not drastically change the layer ranking. Removing EMA leaves the DP mechanism and quantization strategy unchanged but makes layer selection more jittery. We performed an ablation study of ResNet18 on CIFAR10 with and without EMA, and the results are tabulated below.
>
> | count | with EMA | without EMA |
> |-------|----------|-------------|
> | 10    | 56.06    | 54.01       |
> | 15    | 51.59    | 50.07       |
> | 18    | 49.97    | 47.22       |
>
> We see that EMA brings considerable improvement and is effective in smoothing out the noisy sensitivity observations. We will perform a more comprehensive EMA ablation experiment and add it to the revised draft.
>
> Other simple smoothing schemes (e.g., momentum-style moving averages) could play the same role, so our approach is not tied to EMA specifically.
>
> Once again, we thank you for your detailed and helpful review. We hope that our responses and planned revisions have fully addressed your concerns. In light of these clarifications, we would be grateful if you would consider increasing your score.

---

### Official Review · Reviewer_Bdkn · 2025-11-02

**Soundness:** 3
**Presentation:** 3
**Contribution:** 3
**Rating:** 4
**Confidence:** 4

**Summary:**

This paper studies methods to reduce quantization variance in DP-SGD, which slows down the convergence a lot. Technical efforts are dedicated to achieve good performance, maintain DP guarantee, and avoid extra computation. A number of experiments on computer vision and one language modeling are presented on relatively old architectures, showing a near Pareto-optimal utility-compute tradeoffs and about 2X theoretical speed improvement.

**Strengths:**

Overall the paper is clearly written and shows the originality in combining quantization and DP. The problem of DP training efficiency is an important one, highlighting the significance of this work. The experiments are careful and convincing, especially the ablation study. I enjoy reading the details in this paper, such as Pareto optimality and error bars.

**Weaknesses:**

1. Wrong positioning of mixed-precision training

The authors state "While effective for standard SGD, mixed-precision training degrades significantly under DP-SGD. To our knowledge, no prior work has explored mixed-precision training when differential privacy mechanisms are employed." Both sentences are wrong. Mixed-precision training with DP is comparable to DP-SGD with full-precision. See Appendix T of "https://arxiv.org/pdf/2110.05679" (no code though) and Section 3.4/Appendix C of "https://arxiv.org/pdf/2311.11822" (code is open-sourced). The caveat is the loss scaling when using fp16, and DP training with bf16 can just run without loss scaling.

2. Limitation to DP-SGD

I recommend the authors to analyze DP-Adam or DP-SignSGD, where the gradient norm is always \sqrt{d} and line 230 won't hold. How would you extend your analysis in this case? Also, DP-Adam and other adaptive optimizers are the main-stream for large models. Can your method work directly? What will be the performance of quantization empirically?

3. Limitation to training from scratch

DP pre-training is hard. I think this is why the authors stick to small and toy datasets. However, the methodology should be applicable to small-scale finetuning as well. Did the authors try that?

4. What about non-DP?

The methods in Section 5.1 and 5.2 should also improve non-DP quantization, even though additional care was taken to make it DP. Are these methods novel? Are there evidence these methods also work in non-DP scenario?

5. Not practical for throughput

Another minor weakness (and my review score does not depend on this point) is that the improvement of throughput is theoretical, not applicable in practice.

**Questions:**

See weaknesses.

---

> ### Author Response · Authors · 2025-11-24
>
> Thank you for your valuable and detailed feedback on our paper. Below, we address each of your concerns in detail.
>
> ## Incorrect framing of prior mixed precision works.
>
> We thank the reviewer for this correction! Our original wording was indeed too strong. We will revise our text to clarify that existing work studies 16-bit mixed precision, whereas our contribution is to push quantization much further (down to FP4) and to analyze how DP noise interacts with such ultra-low-precision training.
>
> ## Empirical and Theoretical Evaluation for DP-Adam.
>
> We added an experiment for image models trained with DP-Adam. DPQuant outperforms the baseline with a strictly lower privacy budget, especially at 75% and 90% FP4 layers, improving accuracy by ~1 standard deviation:
>
> | model  | count | Baseline | Baseline $\varepsilon$ | DPQuant | DPQuant $\varepsilon$ |
> |--------|:-----:|-------------------|--------------------|--------|---------|
> | CIFAR10| 10    | 54.18 ± 1.48      | 5.05               | 55.03  | 4.97    |
> | | 15    | 49.86 ± 1.57      | 5.05               | 52.40  | 4.97    |
> | | 18    | 47.36 ± 1.36      | 5.05               | 51.18  | 4.97    |
> | GTSRB  | 10    | 72.74 ± 6.04      | 6.05               | 75.87  | 5.91    |
> | | 15    | 58.14 ± 7.64      | 6.05               | 72.32  | 5.91    |
> | | 18    | 50.88 ± 5.11      | 6.05               | 66.45  | 5.91    |
>
> For NLP, our BERT experiment on SNLI already uses DP-AdamW. We apologize for the confusion and will clarify this in Appendix A.4.2 and the main text.
>
> | % quantized | baseline | baseline $\varepsilon$ | DPQuant | DPQuant $\varepsilon$ |
> |-------|------------------------|-------------|-------------------|---------|
> | 0%    | 75.75 ± 0.64           | 7.47        | N/A               | N/A     |
> | 50%   | 62.54 ± 4.54           | 7.48        | 67.80             | 7.48    |
> | 75%   | 52.04 ± 3.95           | 7.48        | 63.61             | 7.48    |
>
> For BERT, DPQuant avoids quantizing the last few layers (including trainable ones) without prior knowledge of layer importance.
>
> Our formal analysis in Section 4 is stated for DP-SGD and relies on the un-preconditioned gradient norm (line 230). In DP-Adam/DP-AdamW, the update is preconditioned so its norm is approximately $\sqrt{d}$, so the argument doesn't apply verbatim. However, the same effect is present.
>
> In DP-Adam, each DP gradient coordinate is $\tilde g_i = g_i + n_i$, where $g_i$ is the true gradient and $n_i \sim \mathcal N(0,\sigma^2)$. The preconditioner divides by a running second moment estimate, so at stationarity the preconditioned coordinate is $u_i \approx (g_i + n_i)/\sqrt{g_i^2 + \sigma^2}$ with $\operatorname{Var}(u_i) = O(1)$. However, the mean (the signal) is $\mathbb E[u_i] = g_i/\sqrt{g_i^2 + \sigma^2} \approx g_i/\sigma$ in the strong-DP regime $\sigma \gg |g_i|$. Thus, DP turns each preconditioned coordinate into unit-variance noise with very small mean. A fixed-precision quantizer on $u_i$ then adds zero-mean noise whose variance doesn't depend on this small mean, so the signal-to-quantization-noise ratio behaves like $g_i^2/\sigma^2$ and deteriorates as privacy noise increases. In the non-DP case ($\sigma \approx 0$), the quantizer sees much larger mean updates and higher SNR, so its effect is mild.
>
> ## Small-scale LM finetuning.
>
> Please see BERT on SNLI results above.
>
> ## Novelty and non-DP applicability.
>
> We agree the mechanisms in Sections 5.1 and 5.2 apply to non-DP settings. However, prior work like LUQ-FP4 shows **FP4 non-DP training suffers negligible degradation**, so DPQuant's improvements would be **very marginal** there.
>
> Our main novelty and motivation are DP-specific:
>
> 1. We identify and analyze a setting where non-DP FP4 training shows negligible degradation, but DP + FP4 suffers substantial accuracy loss, attributing this to the interaction between $\ell_2$-calibrated DP noise and $\ell_\infty$-driven quantization.
> 2. Our loss-sensitivity measures and selection rules are **compatible with DP accounting and composition** (relying only on quantities already DP-released or incorporable into the accountant), which is not a concern in standard non-DP quantization work.
>
> ## Limited end-to-end performance evaluation.
>
> Currently, hardware support for 4-bit quantized training is limited. We included prior works providing FP4 speedup estimates and used conservative estimates.
>
> Prior low-precision training work [1,2] relied on theoretical estimations when hardware and software stacks were immature. With infrastructure availability, realizing our claimed speedups should be straightforward.
>
> [1] Ultra-Low Precision 4-bit Training of Deep Neural Networks, Sun et al.
> [2] Accurate Neural Training with 4-bit Matrix Multiplications at Standard Formats, Chmiel et al
>
> We thank you for your detailed review. We hope our responses and planned revisions have addressed your concerns, and would be grateful if you consider increasing your score.

---

### Official Review · Reviewer_oPEM · 2025-11-02

**Soundness:** 3
**Presentation:** 3
**Contribution:** 3
**Rating:** 6
**Confidence:** 3

**Summary:**

This paper observes that quantization variance is detrimental in DP-SGD. The authors propose two methods to reduce the variance, which achieve good Pareto-optimal performance when used in combination. Specifically, the loss-aware method takes care of DP guarantee. Some experiments on small datasets and models show a good utility-compute tradeoffs while leveraging the quantization speedup.

**Strengths:**

I think the identification of the effects of quantization variance is novel, and the solutions to reduce the variance is novel. Hence this work has fair significance in improving DP efficiency. The motivation of the solutions is reasonable, as variance in DP noise is already known as the key blocker in DP convergence like "Pre-training Differentially Private Models with Limited Public Data". Hence it is not surprising another type of variance from quantization also harms the convergence.

**Weaknesses:**

In practice, especially in NLP, models are optimized by adaptive optimizers like AdamW or Muon, whereas this work has a narrow focus on DP-SGD (maybe not even with momentum?) I would encourage the authors to put DP-AdamW in empirical experiments which should not require extra efforts.

In "Pre-training Differentially Private Models with Limited Public Data", it is claimed that variance only harms the early stage of training, e.g. in pretraining regime, to which this paper's experiments are limited. In fact, many existing works including "Large Language Models Can Be Strong Differentially Private Learners" have shown that DP finetuning is comparable to non-DP finetuning, suggesting variance is less detrimental in the finetuning regime. I wonder would DPQuant be less effective in this regime?

Also please use a different notation for policy. Right now both policy and probability is p.

**Questions:**

Can the authors add DP-AdamW at least for NLP experiments?

Does non-DP quantization works better in finetuning? like comparing fp4 with quantization to fp32 without quantization, will the gap in pretraining be larger than in finetuning?

How necessary is DP quantization in finetuning?

---

> ### Author Response · Authors · 2025-11-23
>
> Thank you for your detailed feedback. We will address each concern below.
>
> ## Applicability to Adam.
>
> We have added an experiment for image models trained with an adaptive optimizer (DP-Adam). DPQuant outperforms the naive baseline while using a strictly lower privacy budget, especially when 75% and 90% of layers run in FP4, where DPQuant improves accuracy by about one standard deviation. The results are:
>
> | model  | count | Baseline | Baseline $\varepsilon$ | DPQuant | DPQuant $\varepsilon$ |
> |--------|:-----:|-------------------|--------------------|--------|---------|
> | CIFAR10| 10    | 54.18 ± 1.48      | 5.05               | 55.03  | 4.97    |
> | CIFAR10| 15    | 49.86 ± 1.57      | 5.05               | 52.40  | 4.97    |
> | CIFAR10| 18    | 47.36 ± 1.36      | 5.05               | 51.18  | 4.97    |
> | GTSRB  | 10    | 72.74 ± 6.04      | 6.05               | 75.87  | 5.91    |
> | GTSRB  | 15    | 58.14 ± 7.64      | 6.05               | 72.32  | 5.91    |
> | GTSRB  | 18    | 50.88 ± 5.11      | 6.05               | 66.45  | 5.91    |
>
> For the NLP setting, our BERT experiment on SNLI is already trained with DP-AdamW as the underlying optimizer. We apologize for not stating this explicitly in the submitted version and will clarify in Appendix A.4.2 and the main text. We restate our key results below:
>
> | count | baseline | baseline $\varepsilon$ | DPQuant | DPQuant $\varepsilon$ |
> |-------|------------------------|-------------|-------------------|---------|
> | 0%    | 75.75 ± 0.64           | 7.47        | N/A               | N/A     |
> | 50%   | 62.54 ± 4.54           | 7.48        | 67.80             | 7.48    |
> | 75%   | 52.04 ± 3.95           | 7.48        | 63.61             | 7.48    |
>
> DPQuant consistently avoids quantizing the last few layers (including trainable ones) without prior knowledge of layer importance.
>
> We will add formal analysis of DP-Adam/DP-AdamW degradation in revision. Intuitively, after preconditioning, DP-Adam updates have coordinates $u_i \approx (g_i + n_i)/\sqrt{g_i^2 + \sigma^2}$ with $O(1)$ variance, but in strong-DP regime the mean shrinks to $\approx g_i/\sigma$ while $\sigma$ is fixed. Fixed-precision quantization adds variance independent of this small mean, so SNR scales like $g_i^2/\sigma^2$ and deteriorates as privacy noise increases. In non-DP case ($\sigma \approx 0$), the quantizer sees larger means and causes less degradation.
>
> ## Applicability to finetuning, including language models.
>
> We appreciate the point raised by the reviewer. Finetuning is an important use case for our work, as differentially private training is most effectively applied onto finetuning workloads. We would like to respond in two parts.
>
> We agree that in prior works such as Zhiqi Bu et al, finetuning workloads are more robust to noise, evidenced by better model performance when trained with pretrained weights. However, as seen in prior works, compared to non-private finetuning, DP finetuning still incurs a penalty on model quality and **is not immune to noise effects**.
>
> * In Figure 1 of Bu et al, while the gap between "AdamW/SGD" and "AdamW/SGD+DP" is smaller in the finetuning case, an accuracy/loss gap remains visible when DP is applied.
> * Table 4 of Bu et al also shows this - when going from non-DP to $\varepsilon=8$, the model shows a decrease in accuracy.
> * In Figure 1b of Li et al, there is a 1% perf gap between $\varepsilon=3$ and $\varepsilon=8$, which is similar to the gap between GPT2 and GP2-medium (which is 3x as large).
>
> **The majority of our experiments are finetuning from pretrained weights.** For image models, we use ImageNet weights for finetuning on CIFAR10 and GTSRB. For the BERT experiment we begin with pretrained “bert-base-cased” weights. For these use cases, we see both (a) a degradation when training with DP and (b) a model degradation when naively selecting layers to train in lower precision. DPQuant targets an **additional, orthogonal source of variance**: low-precision quantization of the already-noisy DP updates.
>
> We demonstrate that even in the “robust” DP finetuning regime, it remains worthwhile to control quantization-induced variance on top of the DP noise.
>
> ## Notation for policy vs. probability.
>
> Apologies for the confusion. We will rename the policy notation to $\pi$ and reserve $p$ solely for probabilities.
>
> ## Necessity of DP quantization in finetuning.
>
> DP quantization is not mathematically necessary. Its role is systems-driven: low-precision formats exploit dedicated hardware (FP4/FP8 tensor cores) and reduce memory/compute costs, crucial for federated and on-device DP finetuning. Our goal is to make low-precision DP viable by avoiding naïve quantization degradation.
>
> Once again, we thank you for your detailed and helpful review. We hope that our responses and planned revisions have fully addressed your concerns. In light of these clarifications, we would be grateful if you would consider increasing your score.

---

### Author Response · Authors · 2025-12-03

We would like to thank the reviewers for their valuable suggestions and comments. Below we summarize how our rebuttal directly addresses the major, recurring comments raised across reviews:
  1. Positioning vs prior work (DP + quantization, improved DP optimizers, pruning/layer selection).
  2. Applicability beyond DP-SGD to Adam/AdamW and other adaptive optimizers.
  3. Effectiveness and necessity of DP quantization in finetuning regimes, including on language models.

For each, we point to specific changes (new experiments, clarified analysis, and revised framing). We hope to provide a more structured view of our rebuttal.

---

comment: **Positioning vs prior work (DP + quantization, improved DP optimizers, pruning/layer selection)** - *reviewers ixjt, Bdkn, ieCQ*
* We expanded the related work section to explicitly discuss:
  * DP + quantization papers (Kang et al., Youn et al., Kim et al., Wang & Başar, Xiong et al.).
  * Improved DP methods (DPAdapter, DPIS, DP-SAT, Denisov et al.).
  * DP pruning / layer-selection (e.g., Differential Privacy Meets Neural Network Pruning).
* We now make the distinction clear:
  * DP + quantization works typically use quantization as the privacy mechanism or a communication/compression tool (often in federated setups), or modify the DP algorithm/parameterization to improve the ε–utility trade-off.
  * **DPQuant instead assumes a fixed dense model and standard DP mechanism, and focuses on making ultra-low-precision (FP4/INT4) training both accurate and hardware-efficient.**
* We highlight the specific gap DPQuant fills:
  * We identify a regime where **non-DP FP4 training is nearly lossless, but DP + FP4 suffers substantial degradation**.
  * We design a **DP-compatible (i.e. privacy preserving), loss-aware layer-selection policy** that directly attacks this DP-specific failure mode, which is not handled in the cited work.

**Together, these changes clarify that DPQuant is not “reinventing” existing methods, but addressing a concrete, DP-specific quantization problem that prior work leaves open.**

---

comment: **Applicability beyond DP-SGD to Adam/AdamW and other optimizers** - *reviewers oPEM, Bdkn, ixjt*
* We added DP-Adam results on CIFAR-10 and GTSRB. Across all configurations, **DPQuant outperforms the naive DP-Adam baseline at a strictly lower privacy budget**, with the largest gains ($\approx 1$ std in accuracy) when 75–90% of layers run in FP4.
* Our BERT–SNLI experiment already uses DP-AdamW. We restated these results and emphasized that **DPQuant improves accuracy at fixed $(\varepsilon,\delta)$** and consistently avoids quantizing the most sensitive final layers.
* We added an intuitive extension of our variance argument to DP-Adam/AdamW: after preconditioning, strong-DP noise yields **small-mean, roughly unit-variance updates**, so **fixed-precision quantization further reduces SNR**; in non-DP settings, larger means make quantization much less harmful.
* We will fold this discussion into the formal analysis in the revision.

**Empirically and analytically, we show that DPQuant is not tied to vanilla DP-SGD and behaves as expected under mainstream adaptive optimizers.**

---

comment: **Effectiveness and necessity of DP quantization in finetuning regimes** - *reviewers oPEM, Bdkn*
* We clarified that **most of our experiments are already finetuning**: ImageNet-pretrained models finetuned on CIFAR-10 and GTSRB, and bert-base-cased finetuned on SNLI, rather than purely “toy” from-scratch training.
* We connected our findings to prior DP finetuning results (e.g., Bu et al., Li et al.), noting that although finetuning is more noise-robust, there remains a **clear DP vs non-DP gap**; in our finetuning setups, **naive FP4 under DP widens this gap**, and **DPQuant substantially closes it** by reducing quantization-induced variance.
* We clarified that **DP quantization is systems-driven rather than mathematically required**: it enables DP finetuning (including language models) to exploit FP4/FP8 tensor cores on accelerators and **low-precision units with lower arithmetic complexity on edge devices**, which is crucial for **federated DP learning** where memory, bandwidth, and compute are constrained.

**For both vision models and language models, DPQuant turns low-precision DP finetuning into a practical option: it preserves the same formal DP guarantees while recovering much of the accuracy that naive FP4 would otherwise lose.**

---

### Meta-Review · Area_Chair_DuHs · 2026-01-07

**Summary:**

The submission proposes a strategy for combining quantization with differentially private SGD optimization.  It works by adaptively selecting a changing subset of layers to quantize at each epoch.

**Reviewer Concerns:**

Reviewers were initially split about whether the submission was marginally above or below the threshold of acceptance.  On the one hand, combining quantization and DP-SGD is an interesting approach, albeit one that has been explored with various focus in a number of papers previously, and the present method seems promising.  On the other, related works were missing, and reviewers wanted more baseline comparisons.

**Reviewer Scores:**

The authors have provided DP-Adam results in the rebuttal, which were requested by several reviewers.  Discussion of specific missed citations has also been added for those raised by Reviewer ixjt.  The main remaining concerns are about the framing of related work as orthogonal or for a different purpose, specifically the concerns of Reviewer ixjt and Reviewer ieCQ.  Additional comparisons to these methods would improve the submission.

---

### Decision · Program_Chairs · 2026-01-26

Accept (Poster)